

# Sharing for science: high-resolution trophic interactions revealed rapidly by social media

Robin A. Maritz and Bryan Maritz

Department of Biodiversity and Conservation Biology, University of the Western Cape, Bellville, South Africa

## ABSTRACT

Discrete, ephemeral natural phenomena with low spatial or temporal predictability are incredibly challenging to study systematically. In ecology, species interactions, which constitute the functional backbone of ecological communities, can be notoriously difficult to characterise especially when taxa are inconspicuous and the interactions of interest (e.g., trophic events) occur infrequently, rapidly, or variably in space and time. Overcoming such issues has historically required significant time and resource investment to collect sufficient data, precluding the answering of many ecological and evolutionary questions. Here, we show the utility of social media for rapidly collecting observations of ephemeral ecological phenomena with low spatial and temporal predictability by using a Facebook group dedicated to collecting predation events involving reptiles and amphibians in sub-Saharan Africa. We collected over 1900 independent feeding observations using Facebook from 2015 to 2019 involving 83 families of predators and 129 families of prey. Feeding events by snakes were particularly well-represented with close to 1,100 feeding observations recorded. Relative to an extensive literature review spanning 226 sources and 138 years, we found that social media has provided snake dietary records faster than ever before in history with prey being identified to a finer taxonomic resolution and showing only modest concordance with the literature due to the number of novel interactions that were detected. Finally, we demonstrate that social media can outperform other citizen science image-based approaches (iNaturalist and Google Images) highlighting the versatility of social media and its ability to function as a citizen science platform.

## INTRODUCTION

Many ecological processes exist as the product of a large number of discrete, ephemeral events. At fine spatial and temporal scales, these events are often difficult to predict, making them challenging to study systematically. This challenge is particularly true for interspecific biological interactions and is magnified when one or both interacting species are difficult to detect, with important impacts on our understanding of the ecology of many systems. Such challenges can be overcome with large investments of time and money, but these costs can be prohibitive and are likely part of the reason for the remarkable absence of empirical

Corresponding author
Robin A. Maritz,
maritzrobin.a@gmail.com

datasets characterising species interactions in ecosystems (*McCann, 2007*; *Hegland et al., 2010*; *Jordano, 2016*).

Together, the origin and development of social media and the concurrent advances in access to mobile cameras represent a disruptive innovation that has changed the manner and rate at which modern events are recorded and communicated. With over 3.80 billion people using social media worldwide (*Kemp, 2020*), the synergy of social media and readily accessible mobile cameras has increased the observational effort of researchers by orders of magnitude. Facebook alone has more than 2.45 billion monthly active users (*Facebook, 2019*) making it the digital platform with the largest social networking potential (*Kemp, 2020*). Harnessing this power has far-reaching implications for understanding ecological and evolutionary processes characterised by difficult to detect, discrete, transient events through the resultant increase in observation coverage and depth.

Trophic interactions, defined as interspecific interactions in which one organism consumes another, form the basis for understanding processes and system characteristics as diverse as energy flow, population dynamics, food web dynamics, and the evolution of behavioural, morphological and physiological adaptations by predators and prey (*Garvey & Whiles, 2017*). Moreover, with a world experiencing climatic changes and worsening environmental conditions (*Vitousek et al., 1997*), attention to species interactions will be crucial for understanding ecosystem function and integrity (*Tylianakis et al., 2008*; *Valiente-Banuet et al., 2014*). Despite their central position in ecological and evolutionary theory, the characterisation of trophic interactions between species and within food webs, particularly those in which such interactions are difficult to study, are often incomplete or absent (*Paine, 1988*; *Chacoff et al., 2011*; *Miranda, Parrini & Dalerum, 2013*; *Jordano, 2016*). Moreover, because certain organismal traits can reduce the detection likelihood of a given trophic interaction, trophic interactions involving terrestrial vertebrates (particularly non-herbivorous interactions) are underrepresented in the literature (*Miranda, Parrini & Dalerum, 2013*). In cases where such datasets exist, endotherms tend to be better represented than ectotherms in trophic studies (*Miranda, Parrini & Dalerum, 2013*)—possibly because of ease of sampling or because endothermy often demands higher food intake rates. Finally, of the interactions reported, organisms involved in lower-trophic-level interactions often suffer from taxonomic aggregation (*Polis, 1991*), which can mask complex species interactions and influence metrics associated with food webs, community assemblages, and interspecific competition (*Greene & Jaksić, 1983*; *Paine, 1988*; *Thompson & Townsend, 2000*). Thus, a method for improving the quantity and quality of collected trophic data is warranted and essential.

Together, reptiles and amphibians (hereafter herpetofauna) include more than 18000 ectothermic, vertebrate species and account for more than half of all global tetrapod diversity (*Pincheira-Donoso et al., 2013*). In many terrestrial ecosystems, these animals can make up a large proportion of the total abundance of vertebrates and contribute significantly to the total biomass of a region (*Western, 1974*; *Iverson, 1982*; *Jacobsen, 1982*; *Petranka & Murray, 2001*). Moreover, herpetofauna (mostly amphibians and squamates) often occupy intermediate trophic levels providing important trophic links between small-bodied invertebrate primary consumers and higher trophic levels occupied primarily by

endothermic predators (e.g., *Polis, 1991*). Interestingly, snakes, a monophyletic lineage of more than 3700 species (approximately 10% of global tetrapod diversity; *Pincheira-Donoso et al., 2013*) are exclusively carnivorous and potentially occupy intermediate trophic positions between many other herpetofauna and species residing at higher trophic levels (*FitzSimons, 1962*; *Greene, 1997*). However, many species of herpetofauna are notoriously difficult to detect and observe in the wild (*Steen, 2010*; *Durso, Willson & Winne, 2011*; *Durso & Seigel, 2015*; *Lardner et al., 2015*; *Rodda et al., 2015*), and individuals of many species feed infrequently or discreetly (*Greene, 1997*), making the systematic observation and quantification of trophic interactions incredibly challenging.

In this article, we demonstrate the utility of a method that uses social media, specifically a dedicated Facebook group, to collect images and videos of difficult to detect feeding interactions involving herpetofauna in sub-Saharan Africa. We hypothesised that information regarding ecological phenomena, specifically trophic interactions, can be collected at large spatial scales and across diverse taxonomic clades by employing the assistance of a large network of potential observers (i.e., Facebook users). First, we highlight the remarkable diversity of predator and prey interactions identified over a five-year period and provide an overview of observer statistics. Next, because snake feeding events are well-represented in our dataset and are notoriously difficult to observe in the wild, we tested the hypothesis that an increased quantity of data on snake feeding would result in the detection of novel species interactions either due to increased sampling effort or differences in detectability. Finally, we compared our dataset to data collected from iNaturalist and Google Images to test whether other digital media platforms offer the observational power required to detect difficult to record trophic interactions. Together, our findings emphasise that previous methods have left gaps in our understanding of feeding interactions involving southern African herpetofauna and to gain a more holistic understanding, data collection must incorporate multiple lines of inquiry. Ultimately, our approach highlights the application of Facebook to rapidly improve trophic interaction sampling coverage and depth in many ecosystems and act as a model for utilising social media to study rare and difficult to detect ecological events.

## MATERIALS & METHODS

### Facebook data collection

The idea for a dedicated predation records Facebook group arose amongst a small group of South African reptile researchers and enthusiasts after noticing feeding observations being shared across the platform and recognising the scientific value in having a place to store and later record these observations. Since then, we have administrated and curated the Predation Records - Reptiles and Frogs (Sub-Saharan Africa) Facebook group (https://www.facebook.com/groups/888525291183325) from its creation in August 2015 until December 2019. We requested that members include details such as predator and prey identity, location, date, time, and observer or photographer's name when sharing an observation to the group. When information was missing, administrators or group members asked for the post to be updated with the necessary details. Predator and prey
identities were confirmed to the finest taxonomic-level possible using a combination of locality information and key physical characteristics and with support from taxon expert group members. In challenging cases, persons with taxon-expertise were consulted using Facebook or via email. Observations that appeared on other social media groups were incorporated in an ad hoc manner.

## Literature data collection

We performed an extensive review of diet records for snake species in southern African snakes (the region where most Facebook observations occurred). We searched primary and grey literature sources (museum bulletins, society newsletters and bulletins, wildlife magazines, and non-indexed journals) for substantiated feeding records. Searches were conducted in English and the main platforms used were Google Scholar and the Biodiversity Heritage Library. Interactions published without supporting details (e.g., field guide descriptions) were categorised as secondary records and were not included in our final analyses. In all instances prey identity was recorded with modification based on updated taxonomy. In instances where only a generic name was provided, the most representative taxonomic name was assigned based on geographic location. Feeding interactions in which multiple prey items of the same type were ingested at once (e.g., 'three nestling chicks') were treated as a single record in the database. Captive-fed observations were recorded but excluded from this study. A list of literature sources used ($N = 226$) and the snake species which they provide data for can be found in the supporting information (Table S1).

## Data management and curation

Data were recorded manually and kept in local storage with monthly back-ups to a personal cloud storage service. The data files are accessible via Figshare (10.6084/m9.figshare. 11920128). Images and videos from all Facebook posts have been downloaded in case users delete or change the privacy settings on their uploaded media. For each feeding interaction, we recorded predator/prey identity, predator/prey life stage, direction of ingestion (for snake predators), interaction specifics (date, time, location), and any noteworthy details. Taxonomic hierarchies were updated automatically for each predator and prey item by referencing a local hierarchy database with information obtained from biodiversity databases (reptile-database.org; sabap2.adu.org.za; amphibiaweb.org; gbif.org).

For Facebook records, additional information included microhabitat (e.g., tree/shrub, artificial surface), type of interaction (true predation or scavenging), type of event (e.g., in situ, roadkill, captured–regurgitated), share date, person who shared the record, person(s) who observed the record, and post permalink. Duplicates were excluded in a semi-automated manner using a photo comparison program (Duplicate Photo Cleaner, v4.7, WebMinds, Inc.). Additionally, records were flagged and verified whenever an identical combination of predator, prey, and observer arose.

For literature records, additional information included predator and prey snout-vent-length, predator and prey mass, type of study (e.g., incidental, museum), museum voucher numbers (when available), and reference. We treated any record in which a given author had published the same interaction previously and did not provide any information on locality or date along with the most recent account as a duplicate.

## Data collection from other digital media sources

We retrieved relevant observations from the iNaturalist citizen science platform (iNaturalist.org) in December 2019. These included all records shared on the iSpot platform (ispot.org.za) for southern Africa that were migrated to the iNaturalist platform during 2017. Currently, there is no centralised method for reporting species interactions on iNaturalist, but a pre-existing iNaturalist project, 'Interactions (s Afr)' (inaturalist.org/projects/interactions-s-afr), gathers feeding interaction data using the observation field "Eating: (Interaction)" which we used to query ("&field:Eating: (Interaction)=") and retrieve all snake feeding records in southern Africa logged onto the platform ($N = 77$). Uncatalogued observations were located using the following independent queries: 'feeding', 'eating', 'meal', 'predation', 'swallow', and 'prey'; species was set to 'Serpentes' and location used was 'southern Africa' ($N = 25$). Records were exported using the download observations function. Duplicate interactions were identified based on iNaturalist observation numbers. The crowd-sourced identification was used when available. We manually inspected images of the target species (the four most observed species in the Facebook dataset) including brown-house snake (*Boaedon capensis*), southern African python (*Python natalensis*), boomslang (*Dispholidus typus*), and cape cobra (*Naja nivea*) for additional instances of feeding that had been missed. Prey could not be identified in nine of the records because the item had already been fully ingested (i.e., food bulge visible).

Google Images results were retrieved in October 2019. Searches were performed for each of the four target study species using the following query: "("*scientific name*" | "*common name*") (eating | prey | predation | swallow | meal | feeding)", and all resulting images were inspected for evidence of feeding. Only images of wild feeding observations were recorded. Photos documenting the same encounter were excluded manually. Observations derived from Google Images were more coarsely identified as the geographic location was frequently missing, but prey were identified to the finest taxonomic-level whenever possible.

## Analytical approaches and comparisons

All data manipulations, graphical outputs, and statistical analyses were conducted in R version 3.6.1. Code can be accessed via Figshare (10.6084/m9.figshare.12287714). In all analyses, non-southern African snake species were excluded. To compare the accumulation rates of Facebook and literature records, the number of records in a given year was averaged with the previous *n* years in 2–15–year windows (moving average). To assess interaction novelty, duplicate interactions within the dataset for each approach were removed. Then, for interactions in which the prey was identified to the species-level (repeated at each taxonomic level) the presence of a given predator–prey interaction was assigned to either literature, digital media source, or both (shared). To test for discordance between the Facebook and literature dataset, Cohen's kappa coefficient ($\kappa$) was calculated using the number of unique and shared interactions (repeated at each taxonomic level) (*Cohen, 1960*). For the comparison of prey-ratios derived from digital media sources, prey items categorised as 'large mammals' are species that typically exceed five kilograms.

## RESULTS

Between 2015 and 2019, we gathered a total of 1917 trophic interactions involving herpetofauna using images and videos shared to the Predation Records - Reptiles and Frogs (Sub-Saharan Africa) Facebook group (Figs. 1A–1L). We detected trophic interactions between 83 families of predators (across 30 orders and 9 classes) and 129 families of prey (across 51 orders and 14 classes) (Figs. 1A, 1G, Fig. S1). Our data encompasses observations from 18 African countries. However, most feeding interactions were observed within South Africa (75.5%; $N = 1446$), which is reflective of a geographic bias in Facebook group participation. Observations were dominated by predation events involving reptiles as the predator representing 66.0% ($N = 1266$) of all trophic interactions in our dataset (Fig. 1A). Remarkably, snakes accounted for the majority of these observations (85.8%; $N = 1086$). We detected feeding events by 85 species of snakes including five of the eight families that occur in Africa.

In our study, there were at least 1369 unique observers who uploaded media documenting a predation event. However, the actual number of observers may be larger because 48 records did not explicitly state who photographed the event. On average, observers shared 1.44 instances of predation (SD = 1.88, range = 32). Across all observations (i.e., any class of predator), 82.0% of observers reported only a single record, whereas, 18.0% of observers reported at least two predation events. Only 0.95% of observers reported more than 10 predation events. There were 891 unique observers who had documented a predation event in which the predator was a snake with an average of 1.24 observations per person (SD = 1.23, range = 19). For instances of snakes as predators, 89.5% of observers had one observation, and only 0.45% of observers have reported more than 10 snake predation events. Notably, the records posted by the top four observers were dominated by observations of prey in road-killed specimens or from snakes that regurgitated prey items while being translocated following a request for the snake to be removed (i.e., to minimise human-wildlife conflict).

Our extensive literature review of southern African snake diets revealed a total of 2884 feeding records covering 109 of southern Africa's 168 species, collected over a period of more than 130 years (Fig. 2). Contrastingly, in five years, we were able to collect 1066 feeding records, which equates to 27.0% of all documented observations. When unsubstantiated records are included, as a conservative measure, our observations from Facebook account for 24.3% of all feeding records.

Overall, feeding observations accrued at a significantly faster rate (Welch's t-test: $t = -3.94, p = 0.0163$) by utilising Facebook ($\mu = 213$ records yr$^{-1}$) compared to historical collection and reporting approaches ($\mu = 20.9$ records yr$^{-1}$). To account for gaps in reporting, we conducted a moving average analysis to test if there were any time periods that produced comparable rates of data accumulation. There were no five-year periods that approached or exceeded the accumulation rate observed using Facebook. The most comparable period was between 2006–2010 ($\mu = 139$ records yr$^{-1}$) which produced 696 records. Only after expanding the time frame to a 10-year window did the number of accumulated records from the literature ($N = 1187$) exceed the number of records that

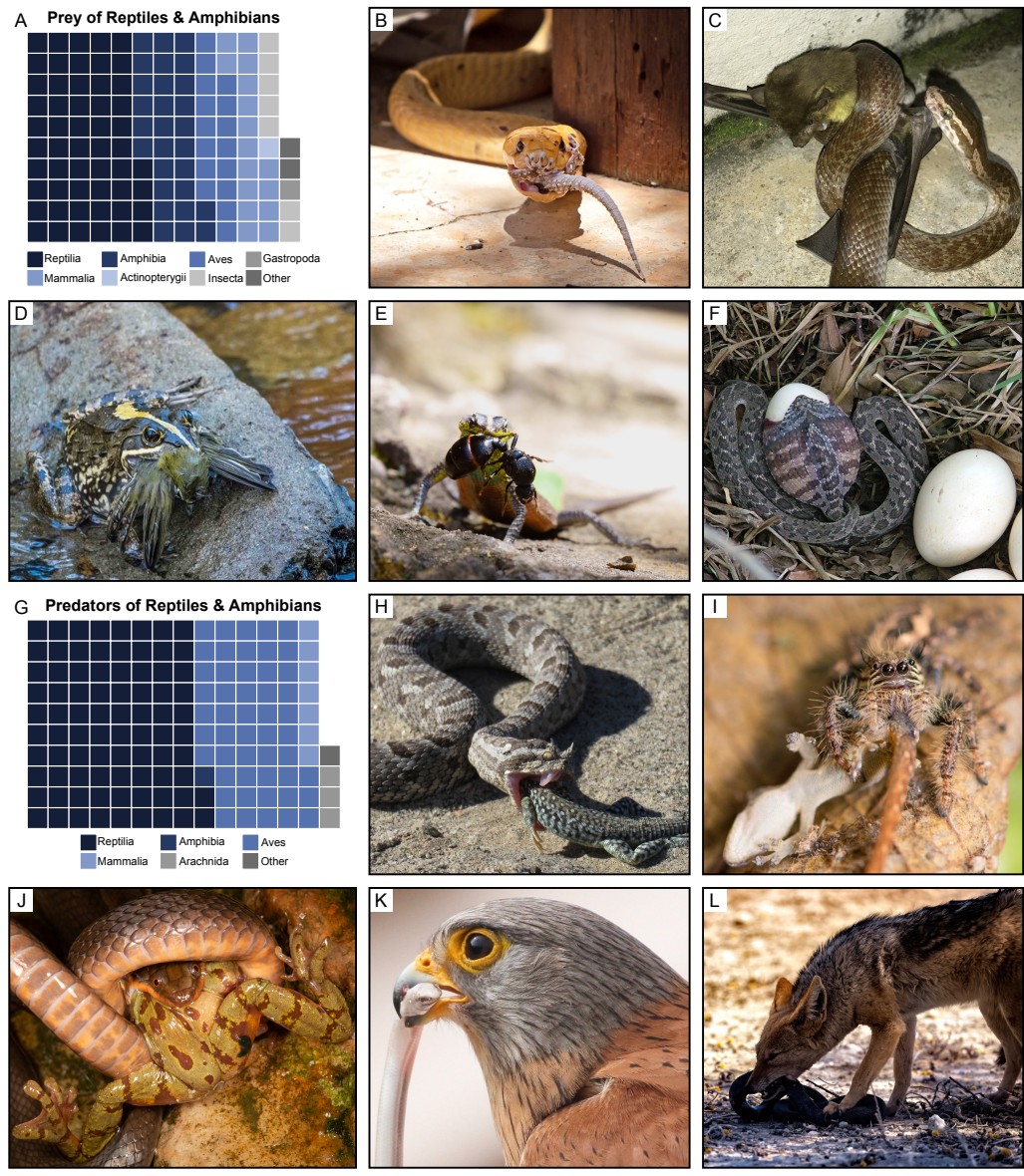

**Figure 1** **Diversity of predator–prey interactions detected using Facebook.** (A) Number of reptile and amphibian feeding records grouped by prey class. Each square represents 10 records. Prey classes with <10 records are included within 'Other' (Arachnida, Chilopoda, Clitellata, Diplopoda, Liliopsida, Magnoliopsida, and Malacostraca). (B–F) Examples of in situ feeding interactions with reptiles and amphibians as predators collected from Facebook. (B) Cape cobra (*Naja nivea*)–Tubercled gecko (*Chondrodactylus* sp.) Photo credit: Michele-Ann Nel. (C) Brown house snake (*Boaedon capensis*)–African yellow bat (*Scotophilus dinganii*) Photo credit: Nick van de Wiel. (D) Cape river frog (*Amietia fuscigula*)–Cape white-eye (*Zosterops virens*) Photo credit: Basil Boer. (E) Waterberg flat lizard (*Platysaurus minor*)–African thief ant (*Carebara vidua*) Photo credit: Kobus Pienaar. (F) Rhombic egg-eater (*Dasypeltis scabra*)–Indian peafowl (*Pavo cristatus*) Photo credit: Sherry Woods. (G) Number of feeding records involving reptiles or amphibians as prey grouped by predator class. Each square represents 10 records.
(continued on next page...)

**Figure 1 (…continued)**
Predator classes with <10 records are included within 'Other' (Actinopterygii, Chilopoda, Insecta, and Malacostraca). (H–L) Examples of in situ feeding interactions involving reptiles and amphibians as prey collected from Facebook. (H) Many-horned adder (*Bitis cornuta*)–Spotted desert lizard (*Meroles suborbitalis*) Photo credit: Jessica Kemper. (I) Jumping spider (Salticidae)–Cape dwarf gecko (*Lygodactylus capensis*) Photo credit: Anton Roberts. (J) Brown water snake (*Lycodonomorphus rufulus*)–Table Mountain ghost frog (*Heleophryne purcelli*) Photo credit: Theo Busschau. (K) Rock kestrel (*Falco rupicolus*)–Bloubergstrand dwarf burrowing skink (*Scelotes montispectus*) Photo credit: Paul Clinton. (L) Black-backed jackal (*Canis mesomelas*)–Mole snake (*Pseudaspis cana*) Photo credit: Reg Lyons.

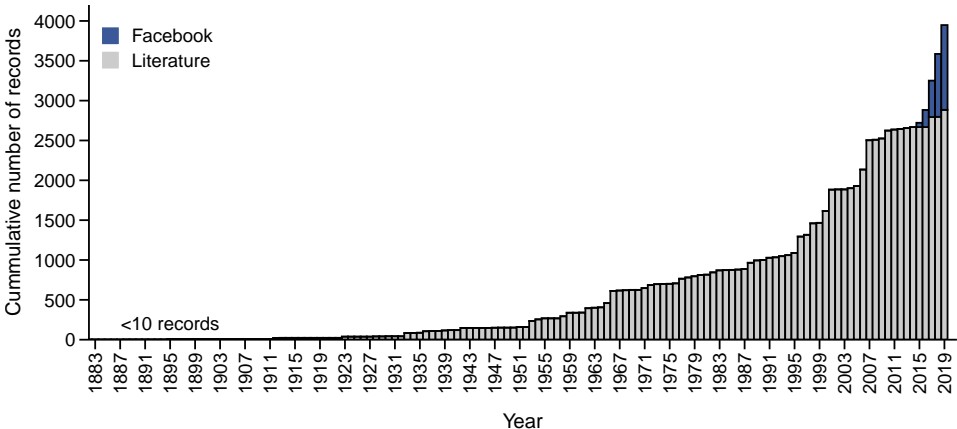

**Figure 2** **Accumulation of snake feeding records from Facebook and literature sources.**

were collected using Facebook in half of the time. This period of high reporting rate can be attributed to the publication of several multi-taxa museum studies from 1998–2007 (*Shine et al., 1998*; *Keogh, Branch & Shine, 2000*; *Webb et al., 2000*; *Webb, Branch & Shine, 2001*; *Shine et al., 2006a*; *Shine et al., 2006b*; *Shine et al., 2007*). Importantly, the periods of comparable reporting rates are the result of decades of work that ultimately culminated in the publication of the literature during those periods, rather than actual rates of record accrual as represented by our Facebook dataset.

We found important differences in the taxonomic resolution to which prey species were identified when comparing the two datasets. Prey were identified to the species level in 76.6% of Facebook records compared to only 50.4% of literature records ($\chi^2 = 216.9$, $p < 0.0001$) (Fig. 3)—probably because digestion of prey items in the gut of museum specimens often eliminates diagnostic characteristics. Similarly, a significantly larger proportion of the Facebook records were identifiable to at least the level of genus, family, and order than records in the literature dataset ($\chi^2 = 49.13$–$250.6$, all $p < 0.0001$) (Fig. 3).

Broadly speaking, the number of feeding observations for each snake species was moderately correlated across the two approaches (Spearman's correlation: $\rho = 0.490$, $p < 0.0001$). However, this relationship obscures some dramatic differences in the overlap in trophic interactions detected via each approach. For interactions with species-level identification of prey items, we identified 441 and 781 distinct interactions within the Facebook and literature datasets, respectively (Fig. 4). Surprisingly, only 114 of these

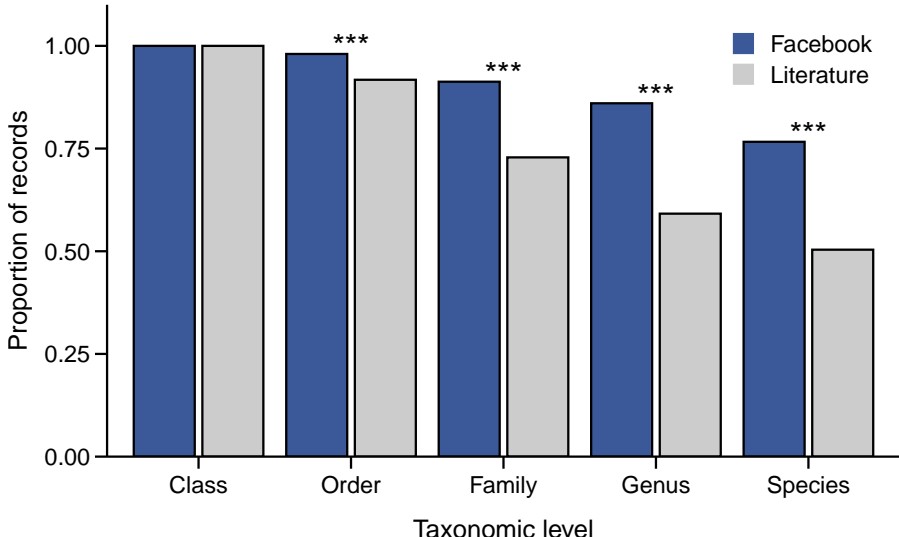

**Figure 3  Taxonomic resolution of snake prey items identified from Facebook and literature sources.**
*** *p*-value < 0.0001.

interactions were shared between the two datasets, and, notably, 327 interactions (of the 441 interactions detected; 74.1%) were unique to the Facebook dataset. Cohen's kappa coefficient ($\kappa = -0.652$) confirmed that the approaches were highly discordant in a non-random manner. Given the bias toward higher-level taxonomic resolution for prey in the literature dataset (Fig. 3), we recalculated Cohen's kappa coefficient with interactions aggregated at the level of genus, family, and order and found low concordance across all taxonomic levels of prey identification ($\kappa = -0.652--0.289$) with order-level taxonomic assignment showing the greatest, but still poor, level of concordance. Depending on level of prey identification (i.e., taxonomic aggregation), our analyses revealed that 28.4–74.1% of the interactions detected via Facebook were previously undocumented (Fig. 4). Remarkably, even at the coarse taxonomic aggregation level of order, 28.4% of interactions detected using Facebook were novel.

On iNaturalist, we found 102 snake feeding observations by querying the database. The earliest upload date of a feeding observations was in 2011 with an average of 11.3 observations added per year since then—a rate that is significantly slower (Welch's t-test: $t = 4.15$, $p = 0.0139$) than Facebook ($\mu = 213$ records $\text{yr}^{-1}$). The greatest number of feeding observations reported on iNaturalist occurred in 2019 ($N = 39$) and represents fewer observations than the number of uploads to our Facebook group during its first year ($N = 54$). Eight of the top-ten snake species recorded in the iNaturalist dataset were also in the top-ten in the Facebook dataset. Notably, the brown house snake (*Boaedon capensis*) had the most observations on both platforms. Interestingly, of the 57 distinct interactions with prey identified to the species-level that were reported on iNaturalist, 19 species interactions were not detected using Facebook and 13 interactions were not present in either the Facebook or the literature dataset.
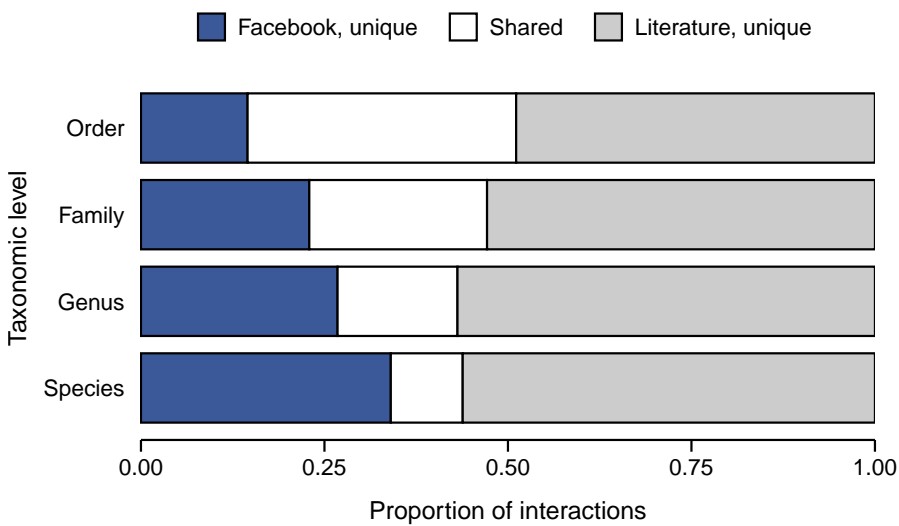

**Figure 4** **Proportion of unique snake feeding interactions identified from Facebook and literature sources.** An interaction is defined as any instance of a specific snake species consuming a specific prey item. Interactions were included only if the prey were identified to the taxonomic level under analysis. Duplicate interactions within an approach were removed.

We gathered an additional seven feeding records for four target species (*B. capensis*, *Python natalensis*, *Dispholidus typus*, and *Naja nivea*) by visually searching through species observations for records that were missed by our querying methods. Across all four target species, Facebook outperformed iNaturalist (Fig. 5A). However, the number of records obtained for each of the target species was proportionally similar across the two platforms. The maximum difference in proportions equated to 3.68% (*D. typus*, Facebook: 7.32% vs. iNaturalist: 11.0%). Finally, at least 9.2% of all iNaturalist records were duplicates of records found using Facebook (i.e., exact photo).

Targeted Google Images searches for the four target species returned 13–25 records per species ($N = 72$) which exceeds the number of records posted to iNaturalist for each target species but still underperformed relative to Facebook (Fig. 5A). 19.4% of records were duplicates of feeding events found using Facebook. Importantly, the ratio of prey types for several of the target species were heavily skewed depending on the source of the observation (Figs. 5B–5E). In particular, 84.2% of records for *N. nivea* depicted ophiophagy (i.e., snake-eating), particularly involving puff adders (*Bitis arietans*) and mole snakes (*Pseudaspis cana*) (Fig. 5E). Additionally, 84.6% of *P. natalensis* observations involved animals feeding on large mammals (e.g., antelope) and our search failed to produce any instances of bird-eating (Fig. 5C). Remarkably, this method did not produce any novel species interactions.

## DISCUSSION

Our study demonstrates the utilisation of Facebook as a crowdsourcing tool to gather a geographically and taxonomically diverse dataset of difficult to observe trophic interactions

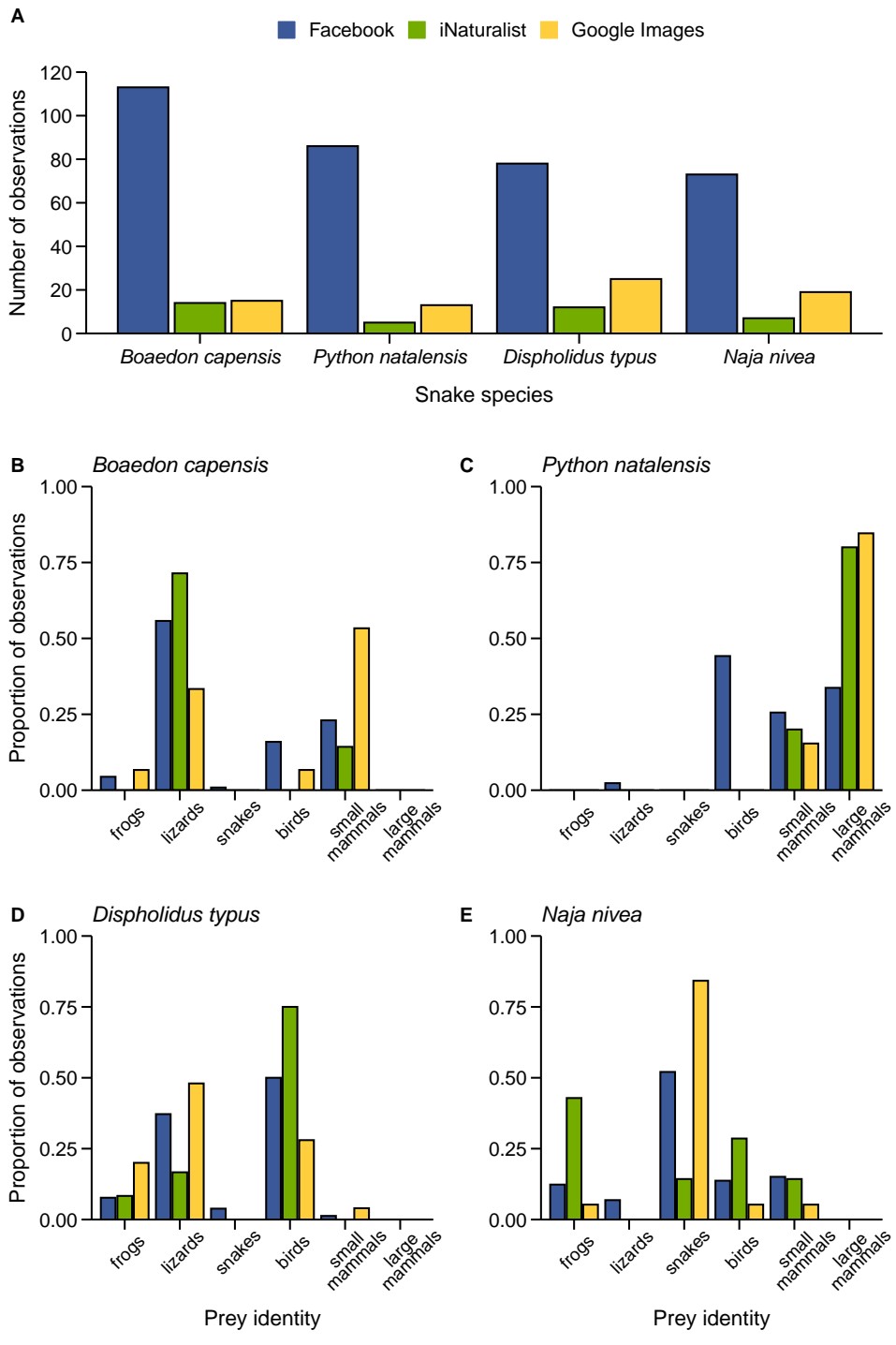

**Figure 5** **Comparison of feeding observations for four target snake species collected from Facebook, iNaturalist, and Google Images.** (A) The number of observations collected. (B–E) The proportion of records with prey belonging to a given prey category.

involving southern African herpetofauna as predators and as prey. Despite these types of interactions being difficult to observe, our approach has yielded observations faster, at finer taxonomic resolution, and that differ significantly from what is currently known within 138 years of herpetological literature. Taken together, these findings provide a powerful example of the potential application of social media to gather discrete, ephemeral ecological interactions.

Importantly, our work is part of a growing recognition of the remarkable power of social media and citizen science to gather biological information (reviewed by *Toivonen et al., 2019* and *Jarić et al., 2020*). Although a number of studies have made use of digital media platforms (i.e., not specifically designed for citizen science) to better understand the geographic and temporal distribution of biological traits or organisms (*Leighton et al., 2016*; *Jiménez-Valverde et al., 2019*; *Marshall & Strine, 2019*), other studies have started to detail ecological and evolutionary processes explicitly. Google Images has been used to quantify insect-pollinator relationships (*Bahlai & Landis, 2016*), commensalism-like relationships between birds and large mammals (*Mikula et al., 2018*), to assess the diets of predatory birds (*Mikula et al., 2016*; *Naude et al., 2019*), and the diets of predatory insects (*Hernandez, Masonick & Weirauch, 2019*). Similarly, Facebook has been used to quantify co-grazing patterns between two deer species (*Mori, Bari & Coraglia, 2018*) and ad hoc observations have revealed a fascinating foraging strategy in skunks (*Pesendorfer, Dickerson & Dragoo, 2018*). Importantly, many of these taxa are often conspicuous due to their size, colouration, microhabitat usage, or duration spent in one location, and the resources in several of the studies are conspicuous (for the same reasons) or spatially restricted. Ultimately, these characteristics improve detection probability and reporting rates. Conversely, our study has demonstrated that social media (specifically Facebook) draws observational power from such a large network that even elusive ecological interactions with low temporal and spatial predictability can be gathered rapidly.

Our approach has several strengths that make its application in ecological and evolutionary studies appealing. Firstly, the ease of reporting means that observers are more likely to share their observations. A dedicated, actively managed, public group allows for photos to be funnelled from across Facebook, and many of our observers had already shared their observations to Facebook in some other context before those posts were shared to our dedicated predation records group. Importantly, the group acts as an outlet for observations that would never otherwise have been documented formally; now, those records can be incorporated into a growing database. Secondly, while citizen science projects like iNaturalist and iSpot attract many users, citizen science platforms are mainly populated by a few very active users (*Sauermann & Franzoni, 2015*). Facebook does not require an inherent interest in a particular topic which allows for a diverse range of media to be posted and shared publicly. Together with the low probability of encountering feeding events—as indicated by the number of single observations in our dataset—dedicated flora and fauna platforms do not attract enough observers to gather sizeable datasets, especially outside of major populated areas. Thirdly, the interactive nature of Facebook facilitates direct communication with observers which can result in more photos or details, if needed. Information such as locality data can be requested directly from observers thus reducing

the reliance on geo-tagging functions of social media platforms, which can be incorrect or missing from posts altogether (*Di Minin, Tenkanen & Toivonen, 2015*). Fourth, the Facebook group format provides an ideal platform to discuss identification of species with interested experts, thereby facilitating expert-crowdsourcing of species identifications. *Austen et al. (2018)* proposed that the identification of species in digital natural history observations should be based on more than one photo and verified by more than one expert. Thus, Facebook groups offer effective mechanisms to meet these criteria. Finally, the community of observers receive informed feedback from researchers regarding their observation. Unlike passive data collection methods (i.e., media and data scraping), active engagement with observers and other members acts as an opportunity to educate the public about the importance of an observation and active engagement and feedback has the potential to incentivise continued participation.

The data gathered via our approach is not without its context-specific challenges. Primarily, our approach does not offer an obvious mechanism for quantifying sampling effort, prohibiting rate- or density-dependent analyses of these processes. Secondly, our approach, as with nearly all sampling approaches, may overrepresent certain interactions in important ways (*Glaudas, Kearney & Alexander, 2017*). For one, our approach is likely to include events that happen (1) frequently, (2) near humans (either urban areas or well-trafficked nature reserves), (3) near humans with the means to access the internet (which shows socioeconomic and regional bias; *Kemp, 2020*), and (4) over longer periods of time. Third, the permanence of posts and their associated media, which appear on social media platforms like Facebook, is not guaranteed, and images may be removed or their visibility settings may be changed by the owner at any time. As a result, there is a need to store images and data outside of the platform in a timely manner. Finally, we have adopted to manually curate and log observations into a database rather than seek automated approaches in part due to the loss of API function in April 2018 associated with a change in Facebook's terms of service (*Freelon, 2018*). This manual approach has worked well at the scale of our analysis but will become problematic at the scale of some of the data that social media has the potential to gather. Advances in machine learning for identification of species in images are progressing rapidly (reviewed by *Wäldchen & Mäder, 2018*) and are starting to be utilised for scientific assessment of social media images (*Di Minin et al., 2018*). However, in our context, we continue to be limited by the fact that the observations being reported are inherently difficult to observe, thus limiting the availability of sufficient amounts of training data. Nonetheless, automation of image identification, or even social media group administration, will be required to scale our approach to truly global ecological or evolutionary questions.

The relatively low measures of concordance between the data gathered via Facebook, and that reported in the literature (Fig. 4), or via other digital media platforms (Figs. 5B–5E) raises the important question of which approach more closely reflects reality. Some approaches to studying diet such as fixed videography (*Glaudas, Kearney & Alexander, 2017*) and DNA barcoding of prey remains (reviewed by *Alberdi et al., 2019*) offer promising future prospects for relatively unbiased dietary analysis for many organisms, including snakes. However, these approaches are incredibly effort- and cost-intensive, limiting their

widespread application. Currently, it is unclear to what degree our data might bias for or against detection of certain interactions. However, we are encouraged by the detection in our Facebook dataset of several apparently difficult to detect interactions (e.g., puff adders (*Bitis arietans*) consuming amphibians, the first reported diet record for Swazi rock snakes (*Inyoka swazicus*)), and interactions with incredibly short handling times (e.g., a vine snake (*Thelotornis capensis*) catching and swallowing a rain frog (*Breviceps* sp.) in under 20 seconds). It is apparent from our analysis that Google Images may be the least effective means for collecting representative diet data, at least for our study system. This is likely to be the case because not all webpages are indexed by Google, (including Facebook) and blogs or media outlets are dominated by eye-catching photos and particularly notable or lengthy encounters. On the other hand, iNaturalist may provide more representative data that can be used in corroboration with Facebook data, which can be promising for geographic regions with more involvement (e.g., United States of America: 483000+; United Kingdom: 28000+; South Africa: 7300+ observers).

Our approach has several implications for our understanding of snake biology. It is well-established that diet has played a major role in the evolution of snakes (*Greene, 1983*; *Colston, Costa & Vitt, 2010*) and their venoms (*Daltry, Wüster & Thorpe, 1996*; *Barlow et al., 2009*; *Casewell et al., 2013*). Additionally, snake feeding, either through demographic effects on prey populations, risk of predation and 'landscape of fear' dynamics, or the selective agents for prey anti-predatory adaptations, are likely to represent the major impacts that snakes have within ecosystems and food webs. Understanding these processes is linked inextricably with high-quality natural history data regarding variation in snake diets, leading to a recent attempt to centralise and analyse feeding data at a global scale (*Grundler, 2020*). However, our understanding of the details of snake diets remains surprisingly superficial, especially in places like Africa where snakebite is a major health concern (*Harrison et al., 2009*; *Chippaux, 2011*; *Murray, Martin & Iwamura, 2020*). In this context, we think that our novel approach to gathering natural history data can provide a powerful tool to supplement existing datasets and, ultimately, improve our understanding of snake feeding, thereby contextualising studies of snakes, their ecological functions, and their venoms.

Our approach has enormous potential beyond our usage of it, and we look forward to seeing its application in multiple ecological and evolutionary contexts. Even within our own dataset, we have only begun to explore the full potential of our data by addressing species-specific questions (*Layloo, Smith & Maritz, 2017*; *Maritz, Alexander & Maritz, 2019*; *Maritz et al., 2019*; *Smith et al., 2019*). However, the dataset lends itself to addressing other questions such as seasonality in feeding and prey preference, intraguild predation, and the evolution of diet. Additionally, we see its value in documenting other ephemeral, discrete, event-driven processes similar to predation, particularly if they can be captured as photographs of the types of subjects that are already shared to social media. For example, photographs of pollinators visiting flowers could be crowd-sourced and curated to better understand pollination dynamics, images of identifiable individual animals (e.g., distinct markings) could be used to assess seasonal body condition, home range size, and lifespan, and photographs of urban biodiversity could elucidate novel urban ecology interactions

between species, or even human-wildlife conflict. Importantly, images of many of these types of events are being shared on social media platforms already, and all that is required is for interested researchers to start engaging with those data.

## CONCLUSIONS

Employing social media as a citizen science platform allowed for the collection of trophic data across a remarkable diversity of interactions involving African reptiles and amphibians. Particularly, the results of the dietary analysis of snakes demonstrate how rapidly and precisely information can be collected to characterise an ecological process compared to traditional approaches. Additionally, the results show a large discordance between sampling via social media and traditional approaches including the detection of many novel interactions, which emphasises how undersampling can lead to gaps in our understanding. Finally, the results highlight how social media can outperform traditional citizen science and crowdsourcing approaches when observations involve elusive animals or unpredictable events, which is likely due differences in the number of active members and thus overall sampling intensity. Beyond herpetological studies, the observational power and approach showcased here has enormous potential for the documentation and investigation of other rare events that underlie important ecological processes, and we emphasise that such approaches should no longer be considered ancillary.

## ACKNOWLEDGEMENTS

We acknowledge Andre Coetzer, Tyrone Ping, and Luke Verburgt for their roles in conceiving the idea for the Predation Records - Reptiles and Frogs (Sub-Saharan Africa) Facebook group. Additionally, we thank them, Gary Nicolau, Nina Perry, and Jason Boyce for assisting in the management of the group's membership and discussions as admins. We are indebted to the members of the group (and other members of the Facebook community) who have shared observations to and participated in our research project. Specifically, we acknowledge the following members for their extraordinary efforts: Helen Badenhorst, Toy Bodbijl, Norman Barrett, Gary Brown, Andre Coetzer, Nick Evans, Kyle Finn, Daniel Karamitsos, Ashley Kemp, Luke Kemp, Trish McGill, Andrea Myburgh, Gary Nicolau, Deon Oosthuizen, Tyrone Ping, Rian Stander, Mike Soroczynski, Francois Theart, and Ryan van Huyssteen. We are grateful for species identifications provided by experts in the community. Particularly, we thank Michael Bates, Alan Channing, Andre Coetzer, Werner Conradie, Adriaan Engelbrecht, Ian Engelbrecht, James Harvey, Teresa Kearney, Luke Kemp, Johan Marais, Gary Nicolau, Dan Parker, Tyrone Ping, Dominic Rollinson, Stephen Spawls, and Luke Verburgt for their input. We recognise the 2017 University of the Western Cape Biodiversity and Conservation Biology Herpetology honours students for assisting in the early phase of the literature search.

### Funding

This work was supported by the National Research Foundation Thuthuka Grant (UID: 118090). The funders had no role in study design, data collection and analysis, decision to publish, or preparation of the manuscript.

### Grant Disclosures

The following grant information was disclosed by the authors:
National Research Foundation Thuthuka Grant: UID: 118090.

### Competing Interests

The authors declare there are no competing interests.

### Author Contributions

- Robin A. Maritz conceived and designed the experiments, performed the experiments, analyzed the data, prepared figures and/or tables, authored or reviewed drafts of the paper, and approved the final draft.
- Bryan Maritz conceived and designed the experiments, authored or reviewed drafts of the paper, and approved the final draft.

### Data Availability

Datasets and R code used in this study are available on Figshare:

Maritz, Robin; Maritz, Bryan (2020): Data for "Sharing for science: High-resolution trophic interactions revealed rapidly by social media". figshare. Dataset. https://doi.org/10.6084/m9.figshare.11920128.v2

Maritz, Robin; Maritz, Bryan (2020): R code for "Sharing for science: High-resolution trophic interactions revealed rapidly by social media". figshare. Software. https://doi.org/10.6084/m9.figshare.12287714.v2.

### Supplemental Information

Supplemental information for this article can be found online at http://dx.doi.org/10.7717/peerj.9485#supplemental-information.

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
