# Peer review of "Sharing for science: high-resolution trophic interactions revealed rapidly by social media"

_PeerJ, doi:10.7717/peerj.9485_

## Round 0.1 · original submission · Minor Revisions

Three reviewers have evaluated your manuscript. All their comments are positive and supportive. They see that your manuscript has a value of novelty and it provides an interesting approach. They see that your manuscript is well written and it is almost suitable to be published, but it still needs some minor modifications.

Please, read carefully all comments provided by our three reviewers and make necessary minor modifications. For example, consider adding some sort of result table as suggested by reviewer #3. You should also improve your discussion chapter a little bit.

Reviewer 1 ·

Basic reporting

Impressive use of social media to obtain data. You have no doubt illustrated the utility of using facebook for your data collection and comparing it to that of iNaturalist and Google images was a bonus.
Your long intro is well organized but your hypotheses at the end are ambiguous. I ran into the same problem with a similar study because I knew the data would prove useful, but struggled to identify specific hypotheses that my results would prove.
I appreciate you clearly stating what your paper will accomplish.
A search for the word hypothesis yielded 1 result, in the lit cited. Simply adding this word to your single, main hypothesis, may help readers and your paper alike.
Your figures are informative, though I would appreciate seeing one that is more visually stimulating and showing an example of a picture that provided good data.

Experimental design

Main research question is ambiguous. Please work on firming that up along with your hypotheses to. It seems that your knowledge gap was the various prey species that were very difficult to detect via traditional data. If so, more focus on that in your hypotheses and conclusions would help.
I know from experience that it is hard to identify a “knowledge gap” with this type of study because social media data is still perceived as new, extra data on top of traditional data. But if we can describe social media data as an important and common dataset that needs to be harnessed regularly, instead of extra data, it may help in identifying new knowledge gaps. Other scientists that do not take advantage of social media data should now be viewed as having a knowledge gap. Thus, part of your knowledge gap should be how to best make use of social media data and you can hypothesize as to the quality and quantity of new data.

Validity of the findings

Your conclusions need to address your main research question, hypothesis, and knowledge gaps. Your conclusions were generally that social media has great utility and provided you with rapid info, catching novel interactions and should be harnessed in other sciences as well. These, to me, are general truths that should be elaborated on. Possibly in another study.

Additional comments

I applaud your use of social media data and encourage you to think of it as a common tool that we should all accept, then move forward in exploring how to incorporate it seamlessly into traditional data sets.

·

Basic reporting

The manuscript proposes a novel approach in the data collection process applied to the study of trophic interactions. The authors showed that the collection of data from a dedicated Facebook group has been capable of supplying data at a higher rate than traditionally used methodologies. This conclusion is supported by the comparison of an extensive analysis of historical data and literature and its comparison with the Facebook data. Moreover, Facebook data allowed us to discover new trophic interactions that have never been previously reported by scientists.

Language is clear and professional. There are no typographical or grammatical mistakes.
Literature reference is sufficient and appropriate. An exhaustive list of the references used for historical analysis has been included in the subscription in the form of an Excel file.

Data collected from Facebook, iNaturalist and Google Image are not attached to the submission. However, it is possible to join the Facebook group from which the data have been retrieved. iNaturalist and Google data could be retrieved since the methodologies used to collect those data are clearly described in the manuscript.

The manuscript provides the correct URL of the Facebook group, but the name cited is wrong. The manuscript states that the Facebook name is “Predation Records - Reptiles & Amphibians (Sub-Saharan Africa)” while it is “Predation Records - Reptiles and Frogs (Sub-Saharan Africa)“. This does not affect the quality of the research as the correct URL of the Facebook group is still provided. However it is an oversight that the authors might want to review.

Experimental design

The original primary research is within Aims and Scope of the journal as the manuscript belongs to the field of biological research.

The paper adopts a meticulous scientific method documenting each step clearly: it is replicable. Moreover, it highlights challenges and respective solutions with motivations behind the choices. Possible limitations of the method are listed and commented on.

Validity of the findings

Conclusions are well stated and supported by the previous part of the manuscript. Data sources are either provided or publicly available.

Additional comments

The methodology proposed is really interesting. It is shown that in certain contexts a group with active users can provide a useful quantitative of data. When thinking about possible applications of the same methodology to other researches, a big challenge can be the creation of a Facebook group focused on the specific topic of the research. How do researchers create such an active and useful group? It would be nice to hear the story behind the birth and growth of the Facebook group used to collect the data analyzed in this manuscript. Did users join the group “spontaneously” or were they close friends or collaborators of the authors? The answer doesn’t affect the validity and soundness of the paper. It is just my opinion that it would be useful and interesting to tell the side story behind the Facebook group so that other scientists can leverage the same methodology for other projects.

First, the paper is sound, clear and exhaustive. Even though there is an abundance of research that uses social media, it is interesting to see an approach solely based on a single Facebook group. The methodology is described in a way that leaves no doubt about its effectiveness in the proposed context.
I believe the paper should be published by PeerJ.

·

Basic reporting

I believe that the manuscript needs at least one table to provide more information about the actual data collected rather than the methodology used. Although is an important work, providing novel and important information, in the manuscript authors are almost exclusively focused on the methodology and the comparison between the sources they used to collect the data. However, they clearly state that several observations were unique and never been reported before in the literature, but those are not reflected in the main text and they should.

Also Figure 1 although is very sophisticated is very hard for a reader and does not provide in a clear way all the information. I suggest a table to be added for supplementing this figure.

Experimental design

this is the strongest part of the manuscript. Very well design and presented.

Validity of the findings

Authors apart from the methodological part they present and the difference between the sources used must also present the actual data and the trophic interactions the detected that are not presented in the literature. Again a table must be really valuable.

Additional comments

Very significant and important work that can be significantly improved if the data collected are well presented.

---

## Round 0.2 · accepted · Accept

Thank you for improving your manuscript according to the review comments and editorial recommendations. You have managed to respond to all minor revision suggestions. I see also that your manuscript is now better than in the previous round. Therefore I see this manuscript is acceptable for publication.